# Altered Expression of Autophagy Biomarkers in Hippocampal Neurons in a Multiple Sclerosis Animal Model

**DOI:** 10.3390/ijms241713225

**Published:** 2023-08-25

**Authors:** Sabrina Ceccariglia, Diego Sibilia, Ornella Parolini, Fabrizio Michetti, Gabriele Di Sante

**Affiliations:** 1Dipartimento di Scienze della Vita e Sanità Pubblica, Università Cattolica del Sacro Cuore, 00168 Rome, Italy; diego.sibilia@unicatt.it (D.S.); ornella.parolini@unicatt.it (O.P.); 2Fondazione Policlinico Universitario “Agostino Gemelli” IRCCS, 00168 Rome, Italy; 3Department of Neuroscience, Università Cattolica del Sacro Cuore, 00168 Rome, Italy; 4Dipartimento di Medicina, Università di LUM, 70010 Casamassima, Italy; 5Istituto di Scienze e Tecnologie Chimiche “Giulio Natta” SCITEC, Centro Nazionale delle Ricerche, 20133 Rome, Italy; 6Dipartimento di Medicina e Chirurgia, Università di Perugia, 06123 Perugia, Italy; gabriele.disante@unipg.it

**Keywords:** autophagy, experimental autoimmune encephalomyelitis, hippocampus, mouse, neurons

## Abstract

Multiple Sclerosis (MS) is a chronic inflammatory disease that affects the brain and spinal cord. Inflammation, demyelination, synaptic alteration, and neuronal loss are hallmarks detectable in MS. Experimental autoimmune encephalomyelitis (EAE) is an animal model widely used to study pathogenic aspects of MS. Autophagy is a process that maintains cell homeostasis by removing abnormal organelles and damaged proteins and is involved both in protective and detrimental effects that have been seen in a variety of human diseases, such as cancer, neurodegenerative diseases, inflammation, and metabolic disorders. This study is aimed at investigating the autophagy signaling pathway through the analysis of the main autophagic proteins including Beclin-1, microtubule-associated protein light chain (LC3, autophagosome marker), and p62 also called sequestosome1 (SQSTM1, substrate of autophagy-mediated degradation) in the hippocampus of EAE-affected mice. The expression levels of Beclin-1, LC3, and p62 and the Akt/mTOR pathway were examined by Western blot experiments. In EAE mice, compared to control animals, significant reductions of expression levels were detectable for Beclin-1 and LC3 II (indicating the reduction of autophagosomes), and p62 (suggesting that autophagic flux increased). In parallel, molecular analysis detected the deregulation of the Akt/mTOR signaling. Immunofluorescence double-labeling images showed co-localization of NeuN (neuronal nuclear marker) and Beclin-1, LC3, and p62 throughout the CA1 and CA3 hippocampal subfields. Taken together, these data demonstrate that activation of autophagy occurs in the neurons of the hippocampus in this experimental model.

## 1. Introduction

Multiple Sclerosis (MS) is a neurological disorder involving complex autoimmune mechanisms and targeting the central nervous system (CNS). In MS, the main pathological hallmarks include chronic inflammation, demyelination, neuro-axonal damage, and gliosis [1,2]. Several studies have demonstrated that also neurodegeneration participates in the pathogenesis of MS, contributing to brain tissue atrophy with consequent neurological disability, physical deficits, and a global cognitive impairment [3]. The hippocampal region is well known for its key role in learning and memory activities. Alterations of cellular mechanisms in the hippocampus have been described, in vivo and in vitro, in distinct neurodegenerative disorders characterized by cognitive and behavioral dysregulation [4,5,6,7,8].

Experimental autoimmune encephalomyelitis (EAE) is an animal model extensively used to investigate pathogenic mechanisms of MS. Indeed, the EAE murine models display many of the pathological features that characterize MS disease including neuroinflammation and neurodegeneration in different areas of CNS [3]. Clinical and experimental studies have well-documented the involvement of the hippocampus during MS and its experimental model EAE. In MS patients, analysis performed through neuroimaging, such as magnetic resonance, documented that hippocampal damage is correlated to learning and memory difficulty along with depressive symptoms [9]. The functional alterations of the hippocampal activity have been reported to be due not only to tissue atrophy but also to the disconnection of this region from several brain networks [9]. Hippocampal atrophy, demyelination, reduction of synaptic connections, loss of interneurons in the CA1 subfield, and activation of microglia and astrocytes have been shown in chronic EAE [10,11,12,13,14]. The cellular mechanisms occurring in the hippocampal neurons and in charge of the cognitive deficit(s) during EAE still remain to be clarified. 

Autophagy is a cellular process that maintains CNS homeostasis by degrading and recycling protein aggregates, damaged organelles, and microorganisms and improves cell survival in stress conditions. In macro-autophagy, herein referred to as autophagy, cytoplasmic contents are sequestered within double-membrane vesicles, the autophagosomes, which subsequently fuse with the lysosomes, forming autolysosomes that mediate the degradation of substrates, recycling of cellular material and energy renewing [15]. Autophagy is involved in many physio-pathological processes including embryogenesis, cell differentiation [16], neurodegeneration [8,17], cancer [18,19], bacterial and viral infections [20,21,22,23], and other types of stress [24]. Several studies documented that, in the pathogenesis of MS and its experimental model EAE, autophagy plays a critical role in distinct types of tissues and cells [25,26]. In MS, autophagy participates in the survival and differentiation of oligodendrocytes, and their myelinating capacity [27]. The upregulation of autophagy in these cells enhances their ability to increase the thickness of the myelin sheaths as well as the number of myelinated axons [28]. The level of expression of the Atg5 protein, an autophagic regulatory marker, is increased in the T-cells of blood and brain from MS patients, suggesting a possible involvement of autophagy in the activation of T-cells (which, when activated, contribute to destroy myelin sheaths of axons) [29,30]. EAE mice showed defective autophagy in the neurons of the spinal cord, and pharmacological induction of this process with rapamycin, reduced inflammation, demyelination, and neuronal loss [31,32]. On the contrary, inhibition of autophagy with 3-methyladenine resulted in higher neuronal apoptosis in EAE mice [31], suggesting that autophagy dysfunction could be associated with EAE-induced neuronal loss.

To date, autophagy involvement in hippocampal activity and in the dysfunction of this region in EAE mice is still poorly understood. 

The present work analyzes, for the first time in the hippocampi of mice affected by EAE, the autophagic process through the evaluation of the main autophagy-related proteins including Beclin-1, microtubule-associated protein light chain (LC3), and p62, also called sequestosome1 (SQSTM1). In this respect, also, the protein kinase B/mammalian rapamycin target protein (Akt/mTOR) [33] has been investigated to demonstrate whether autophagy activation or dysfunction occurs in this brain region. Our data showed that Beclin-1, LC3 II (LC3 I linked to phosphatidylethanolamine (PE)), and p62 are localized, exclusively, in the neuronal cells, and that their expression levels were reduced in the hippocampi of mice affected by EAE; as well, also Akt/mTOR pathway was significantly downregulated. Interestingly, taken together, all these results indicate an upregulation of autophagy in the EAE-derived hippocampal neurons.

## 2. Results

### 2.1. The Beclin-1, LC3 II, and p62 Expression Levels Decreased in the Hippocampus of EAE Mice

To monitor the autophagy, we investigated the classical markers of this process by evaluating the expression levels of the Beclin-1, LC3 II, and p62 through Western blot analyses of hippocampal tissue homogenates of control and EAE mice. 

Beclin-1 is an autophagic marker that regulates the stage of formation of double-membrane vesicles, the autophagosomes, and the level of LC3 II expression [34]. Our experiments showed that the Beclin-1 level is significantly reduced (*p* < 0.05) in EAE samples compared with the control animal group (Figure 1a).

LC3 is a protein that contributes to regulating the initial stages of the autophagic process. It is cleaved in LC3 I, which represents the cytosolic constitutive form. LC3 I then is conjugated to PE to constitute LC3 II active form. LC3 II is present both on the inner and outer membrane of the autophagosome, and its amount is correlated to the number of vacuoles inside the cells [35]. Western blot assay showed that, comparing EAE and control samples, the expression levels of LC3 I were similar (bands corresponding to 16 kDa) (Figure 1b), while LC3 II levels were significantly reduced (*p* < 0.05, bands corresponding to 14 kDa) (Figure 1c). In addition, the amount of LC3 I that we detected was more abundant (Figure 1b) and, therefore, more easily detectable than LC3 II (Figure 1c) in the hippocampus. This result was in line with the observation of Mizushima [36] showing that in the mouse brain, the constitutive form (LC3 I) is rather more expressed than the active one (LC3 II), both in the physiological and pathological conditions.

The p62 protein binds the ubiquitinated protein aggregates to promote their degradation through the autophagic process inside autolysosomes [37]. Our results showed that the expression level of p62 is significantly lower (*p* < 0.001) in the EAE group compared with controls (Figure 1d). 

### 2.2. Beclin-1, LC3, and p62 Were Localized in Hippocampal Neurons of Control and EAE Mice

To analyze the distribution of the autophagy-related proteins in the hippocampal regions and to identify the cellular type/s in which they are expressed, double-labeling immunofluorescence experiments were performed on brain sections of control and EAE mice. We used specific antibodies against Beclin-1, LC3 (which does not distinguish LC3 I from LC3 II form), and p62 in combination with a variety of cell-specific markers, such as neuronal nuclear (NeuN) protein for neurons, glial acidic fibrillary protein (GFAP) for astrocytes and CD68 for activated microglia. We detected the co-localization of each autophagic protein, Beclin-1 (Figure 2), LC3 (Figure 3), and p62 (Figure 4) (red), with NeuN marker (green) in the cytoplasm of most neurons in CA1 (Figure 2c,f, Figure 3c,f and Figure 4c,f) and in some CA3 neurons (Figure 2i,l, Figure 3i,l and Figure 4i,l) both in the control and EAE animals. Interestingly, the immunofluorescent signals of Beclin-1 and LC3 were both localized in the soma and along the fibers of most of the neurons in CA1 (Figure 2b,e and Figure 3b,e) and in the soma of some neurons in CA3 (Figure 2h,k and Figure 3h,k). The p62 labeling was evident in the cell body of most of the neurons in CA1 (Figure 4b,e) and in some CA3 neurons (Figure 4h,k). According to our Western blot data, the immunofluorescence images confirmed that the immunoreactivity of all the analyzed autophagy proteins (red) was visibly more marked and diffuse in hippocampal neurons of control mice (Figure 2b,h, Figure 3b,h and Figure 4b,h) compared with EAE animals (Figure 2e,k, Figure 3e,k and Figure 4e,k).

For interpretation of the references to color in this figure legend, the reader is referred to the Web version of this article.

To confirm that the immunoreactivity of LC3 (and thus also of Beclin-1 and p62) was localized only in neurons, double-label immunofluorescence experiments, using antibodies against GFAP or CD68 and LC3 were performed (Figure 5). As expected, the LC3 marker (red) localizes neither in the astrocytes (green) nor in activated microglia (green) in CA1 (Figure 5a,b,e,f) nor in CA3 (Figure 5c,d,g,h) areas, in control and EAE mice. Noteworthy, as already known [13], the GFAP and CD68 reactivity markedly increased in the hippocampus of EAE animals (Figure 5b,d,f,h), compared with controls (Figure 5a,c,e,g).

No evident labeling for all autophagic markers analyzed was observed in the different cell types of the CA4 subfield.

### 2.3. Akt/mTOR Pathway Is Not Activated in the Hippocampus of the EAE Mice

The Akt/mTOR signaling pathway plays a significant role in the regulation of a wide variety of biological processes including autophagy [38]. In the brain, and, particularly in the hippocampus, mTOR signaling plays a critical role in modulating physiological functions, including synaptic plasticity, memory storage, and cognition [39]. Detection of the phosphorylation state of Akt/mTOR allowed for the measurement of the level of the activation of the proteins involved in this relevant pathway. In this respect, Western blot assay was performed in control and EAE animals to evaluate both the Akt and mTOR total levels and Akt (Ser473) (p-Akt) [40] and mTOR (Ser2448) (p-mTOR) [41] phosphorylated levels. As shown in Figure 6a,b, the ratio values of p-Akt/Akt and p-mTOR/mTOR significantly decreased (respectively *p* < 0.01 and *p* < 0.05) in the hippocampus of EAE mice compared to the control values. Noteworthy, the p-mTOR levels in control animals were markedly expressed. 

## 3. Discussion

In the present study, we investigated, for the first time, the expression levels, distribution, and cellular localization of the main autophagy-related proteins including Beclin-1, LC3, and p62 in the hippocampus of EAE-affected mice. Our data showed that Beclin-1, LC3, and p62 expression were reduced after EAE induction and, that all these markers were localized exclusively in hippocampal neurons of CA1 and CA3 areas. In addition, Western blot analysis evidenced a significant downregulation of the Akt/mTOR signaling pathway after EAE. Collectively, the results obtained are consistent with an enhancement of the basal levels of autophagy in the hippocampal neurons of EAE animals compared to controls. The study of autophagy biomarkers in the hippocampus of mice affected by a recognized model of MS (EAE) constitutes an element of novelty, which is in fact present in our work.

It is well known that cognitive dysfunction is one of the disabling hallmarks of the MS disease [42]. The involvement of the hippocampus in the cognitive deficit due to demyelination, synaptic alterations, neuronal loss, and correlated learning-memory impairment has been widely demonstrated in EAE, the animal model of MS [10,14,43,44,45,46]. However, the cellular and molecular mechanisms involved in hippocampus damage, have not yet been fully investigated and elucidated in EAE mice. A better understanding of autophagic stress and further identification of autophagic cell mechanisms may lead to therapeutic interventions that help restore homeostasis in the neurons in EAE and MS.

Autophagy is a cellular process also known to enhance synaptic plasticity and play a significant role in improving memory in hippocampal neurons [47]. To date, there are no studies regarding autophagy regulation in the hippocampus of EAE mice. Therefore, the important goal of this work is to evaluate autophagic activity through the analysis of the main key proteins that regulate this process. At first, we investigated Beclin-1, one of the upstream regulatory proteins of the autophagic pathways. Indeed, Beclin-1 is a protein required for the induction of autophagy, for its role in the regulation of the formation and maturation of new autophagosomes [34]. In the present work, we showed a significant reduction in Beclin-1 expression levels in the hippocampus of EAE mice compared to controls (Figure 1a). Subsequently, Western blot experiments have shown a significant reduction of LC3 II levels in EAE-affected animals compared to the control group (Figure 1c). Since LC3 II is a protein associated with the autophagosome membrane, its reduced expression is related to a reduced formation of autophagosomes. The low levels of Beclin-1, a protein known to regulate LC3 II synthesis, may in turn be the cause of the decreased LC3 II expression and, of the limited formation of autophagosomes in our model, as indicated also by Kang [34] in a previous study. In addition, it is known that the amount of LC3 II levels is directly proportional to the activation of autophagy, declining in the case of prolonged activation of autophagic processes due to autolysosomal degradation [48]. In this respect, the reduced levels of LC3 II that we detected, due to its degradation, suggest either decreased formation of autophagosomes or, consequently, an increased autophagic flux in hippocampal neurons of EAE mice. Since the activation of LC3 I to LC3 II may be necessary but not sufficient to trigger cellular autophagy, one of the most widely used methods to monitor autophagic flux is based on changes in levels of p62 expression. Indeed, the p62 protein is a receptor for cargo expected to be degraded by autophagic activity. Furthermore, p62 constitutes a selective substrate of autophagy as it binds LC3 for incorporation into autophagosomes [49]. Therefore, p62 accumulates when autophagic activity is inhibited and its levels are low when autophagy is activated. [37,50]. In our EAE model, Western blot experiments showed a significant decrease in the p62 expression levels during EAE in comparison with controls (Figure 1d), indicating that the fusion autophagosome-lysosome occurs to form the autolysosomes, where the cargo is degraded, and, therefore, autophagy is activated. On the contrary, previous studies, performed in the spinal cord of a murine model of EAE, have demonstrated defective autophagy characterized by decreased levels of Beclin-1 and LC3 II and increased expression of p62, that indicated an impairment in the fusion between autophagosomes and lysosomes [31,32].

It is important to highlight that the basic autophagic proteins, Beclin-1, LC3, and p62, were analyzed in the same context of the EAE/MS murine model but in different regions of the CNS, as the hippocampus in our study and the spinal cord in the research by Feng and co-authors, 2017, contribute differently to the activation and, consequently, to the role of autophagy in damaged neurons.

Altogether, these data innovatively indicate that autophagy is activated in the hippocampus of EAE mice, and future studies on the modulation of these analyzed markers in the hippocampus could have a therapeutic potential in memory and learning impairment in EAE-affected mice. Interestingly, it is also known that endoplasmic reticulum (ER) stress occurs in the hippocampi of EAE mice [51] and autophagy is closely interconnected with ER stress, including ER-stress-mediated autophagy activation and the formation of autophagosomes at the ER membrane [52]. In this regard, further studies could help to deepen and better understand the role of ER stress associated with autophagy activation that we have observed in the hippocampus of EAE mice.

In addition, double immunolabeling analysis evidenced that the Beclin-1, LC3, and p62 proteins are all exclusively expressed in neuronal cells of the CA1 and CA3 subregions both in control and EAE mice (Figure 2, Figure 3 and Figure 4). Interestingly, we observed that Beclin-1 and LC3 labeling is localized not only within cell bodies but also along fibers of the neurons of the CA1 region. Autophagosome precursors are known to form in distal axons and then, during maturation, to move, retrogradely, along the axons, toward the soma [53,54]. 

To improve the understanding of neuronal autophagy and its precise regulation is important for developing novel strategies to treat MS disease. The activation of autophagy localized in soma and/or axons may represent a beneficial therapeutic intervention both for the EAE model and for MS, even though the potential contribution of modulated autophagy must be carefully monitored.

Since our results indicate a loss in the number of mature autophagosomes, it will be interesting to perform a study to verify whether the process of demyelination that occurs in MS/EAE may alter the axonal transport of autophagosomes and their maturation. Immunofluorescence images showed evident reactive astrogliosis and activation of microglia localized in the CA1 and CA3 areas of EAE animals (Figure 5), previously demonstrated by other authors [13]. It is interesting to underline that the CA1 and CA3 of the hippocampus involved in the expression of the autophagic markers are also the areas involved in hippocampal atrophy and its functional impairment. In a previous study, it has been observed that these regions showed a reduction of volume in MS patients and, evident demyelination and loss in the number of neurons in EAE mice [10].

Next, we have investigated the involvement of the Akt/mTOR signaling pathway in autophagy regulation. The Akt/mTOR signaling is known to promote cellular growth, differentiation, and survival in physiological conditions and in response to environmental stress [55]. Deregulation of this pathway has been reported in human diseases, including diabetes, neurodegenerative diseases, and cancer [56]. mTOR is not only a downstream target of the Akt pathway but it is also involved in the regulation of autophagy [57]. Furthermore, mTOR is a key regulator of physiological functions, including synaptic plasticity, memory storage, and cognition in the hippocampus [39]. Precedent studies have demonstrated that the deregulation of mTOR signaling participates in cognitive impairment by modulating autophagy [58,59,60]. Thus, we have examined the activity of Akt/mTOR in the hippocampus of control and EAE mice. We demonstrated that, after EAE treatment, the levels of p-Akt and p-mTOR were significantly decreased (Figure 6a,b), indicating that autophagy increased as pathway Akt/mTOR is downregulated [33]. These results have pleasantly surprised and intrigued us representing an element of novelty since, in previous studies of EAE models [32,61] and in other disorders [62], generally, the authors demonstrated elevated expression levels of these signaling proteins. 

These data provide a molecular basis to modulate therapeutically autophagy in neurons. It is important to identify signaling pathways involved, such as Akt/mTOR, and determine how they can be regulated in vivo. The availability of small molecule effectors (inhibitors and activators) will allow the determination of whether autophagy can be modulated to make neurons more resilient to the cellular stressors of the disease, offering some new tools for the future in ameliorating cognitive impairment in MS.

In summary, in this work, we demonstrated, for the first time, that autophagy is markedly activated in the hippocampal neurons of EAE-affected mice since the formation of autophagosomes is reduced and the concentration of p62 levels decreased. In addition, the Akt/mTOR signaling pathway is not activated consistently with increased autophagy activity observed. 

Essentially, through this study, we wanted to provide a basis for elucidation of autophagy activation in the hippocampus, which may constitute an important player in MS processes, which appears not to be adequately investigated, and we started, necessarily, from the analysis of the primary molecular mechanisms, which constitute a prerequisite for further studies more deeply investigating these processes. However, future additional research will be needed to address the precise mechanism/s of autophagy in hippocampal neurons essential for the identification of therapeutic targets, to investigate whether and how autophagy might be modulated to influence the neurobiological and neuropathological features of EAE and MS, and to build new therapeutic approaches. 

## 4. Materials and Methods

### 4.1. EAE Induction and Monitoring

The induction of EAE has been performed by immunizing C57Bl/6 mice (female, *n* = 8) subcutaneously (day 0) with an emulsion of an adjuvant, derived from Mycobacterium Tuberculosis (4X-concentrated complete Freund’s adjuvant, CFA, 200 μg/mouse) and a myelin antigen, the MOG35-50 (50 μg/mouse). In addition, Bordetella Pertussis toxin (BDT, total 300 ng/mouse) has been administered intraperitoneally (i.p., days 0 and 3). All the above-mentioned reagents were purchased from Sigma-Aldrich, Merck, Germany. SJL/J mice were randomly distributed into two distinct groups: the EAE group (*n* = 8), immunized with the above-described emulsion and boosted with BDT, and healthy controls (*n* = 8), immunized with an emulsion of CFA-4X and phosphate buffer saline (PBS) and boosted with the same amount of PBS. A daily monitoring of body weight and of the development of clinical signs and symptoms was performed, following the score scale described in previous works [14,63,64,65]. Mice were monitored daily for 22 days, waiting for the development of clinical signs of the acute phase, and sacrificed when their score comprised between 2 and 3, according to the procedures of previous works [66,67,68,69] (Figure 7). We adopted the following score criteria: 0.5 = unsteady gait or weak tail (mouse walks on the bench like a duck or weak tonus or half of tail drops); 1 = weak tail and hindleg weakness/unsteady gait or limp tail (mouse walks on the bench like a duck and tail drops or weak tail); 1.5 = Unilateral weakness of hindlimb (while walking on the grid of a cage one leg of the mouse falls into the gaps); 2 = Bilateral weakness of hindlimb (while walking on the bench one hindlimb is not moving or both legs fall into the gaps of grid or combination of previous scores); 2.5 = Unilateral hindlimb paraplegia (mouse walks on the bench and one hindlimb is not moving; mouse walks on the grid of the cage and one hindlimb is always in the gaps); 3 = Bilateral hindlimb paraplegia (mouse walks on the grid of the cage and both hindlimbs are always in the gaps; mouse walks on the bench and is not moving its hindlimbs; lower body control has to be checked). The score evaluation was performed only in the morning and assessed by two different operators (blind). Results were then registered by a third operator from the mean calculated by each different visiting operator for each day, except for the weekend.

### 4.2. Ethical Considerations

The breeding, backcrossing, and maintenance of mice were performed at the animal facility of Catholic University, CenRis, Rome. All experimental work has been conducted in accordance with relevant national legislation on the use of animals for research, referring to the Code of Practice for the Housing and Care of Animals Used in Scientific Procedures and the protocol was approved by the Ethics Committee of animal welfare organization (OPBA) of the Catholic University of Rome and by the Italian Ministry of Health (authorization 15/2021-PR protocol 1F295.120), whereby mice with premature disease severity and overt suffering were excluded and sacrificed before the end of the timepoint (*n* = 2). Every effort was made to minimize the number of animals used. 

### 4.3. Western Blot Experiments

Mice (control and EAE, 5 animals/group) were sacrificed by decapitation after deep i.p. anesthesia with 87.5 mg/Kg body weight of Ketamine (Merck Sharp & Dohme, Rahway, NJ, USA) and 12.5 mg/Kg body weight of Xylazine (Bayer, Leverkusen, Germany), 0.1 mL/20 g body weight, for the Western blot analyses. The brains were removed, and the hippocampus was rapidly isolated on ice from the ipsilateral hemisphere and stored at −80 °C before being processed. Subsequently, the samples were mechanically homogenized in an ice-cold lysis buffer containing 50 mM Tris-HCL, pH 7.4, 150 mM NaCl, 1% Triton X-100, 0.1% sodium dodecyl sulfate (SDS), 0.5% sodium deoxycholate, and with a protease and phosphatase inhibitor cocktail (Sigma-Aldrich, St. Louis, MO, USA) added just before use. The lysates were incubated on ice for 30 min, sonicated, and then centrifuged at 14,000× *g* for 30 min at 4 °C. The supernatants were used to detect the expression of Beclin-1, LC3, p62, Akt, p-AKT, mTOR, and p-mTOR using appropriate antibodies (all purchased by Cell Signaling Technology, Inc., Leiden, The Netherlands). The supernatants were collected, and protein concentrations were determined using the BCA Protein Assay Kit (Thermo Fisher Scientific, Rockford, IL, USA) [70]. Equal amounts of proteins (25 μg/lane) from each sample were separated by SDS-PAGE in a 4–20% precast polyacrylamide gel (Bio-Rad, Hercules, CA, USA), and blotted to polyvinylidene fluoride (PVDF) membranes (Bio-Rad, Hercules, CA, USA) using the Trans-Blot Turbo transfer system (Bio-Rad, Hercules, CA, USA) at 1.3 A, 25 V for 7 min (5 min for LC3, and 10 min for p-mTOR). Non-fat dried milk (5% *w*/*v*) in Tris-buffered saline (50 mM Tris-HCL, pH 7.4, 400 mM NaCl, 0.1% Tween 20) (TBS-T) was used to block the membranes for 60 min at room temperature (RT). The membranes were then incubated overnight at 4 °C with the appropriate primary antibodies diluted 1:1000 in 5% non-fat dried milk in TBS-T, except β-Actin (Sigma-Aldrich, St. Louis, MO, USA), which has been used as normalizer at 1:5000. After three washes with TBS-T, PVDF membranes were incubated for 2 h at RT with adequate horseradish peroxidase-conjugated secondary antibody IgG (1:5000, Invitrogen, Carlsbad, CA, USA) diluted in 5% non-fat dried milk in TBS-T. After three washes, membranes were incubated with ECL solution (Bio-Rad, Hercules, CA, USA) and bands were visualized using the Chemidoc Imaging System (Biorad, Hercules, CA, USA). Densitometry analysis was performed using ImageJ software, versione 1.53k. 

All Western blot analyses were performed in at least three independent experiments. 

### 4.4. Immunofluorescence Cellular Localization of Autophagic Markers

The mice (control and EAE, 3 animals/group), after anesthesia, were perfused transcardially with 0.9% saline followed by 4% paraformaldehyde (PFA). The brains were removed, post-fixed in 4% PFA overnight at 4 °C, and washed in phosphate buffer (PB), 0.1 M, pH 7.4. Sections were cut on a vibratome (Leica VT 1000S) at a thickness of 40 µm and collected in PB. Nonspecific binding sites were blocked with 5% normal donkey serum in PBS 0.01 M, pH 7.4, for 60 min at RT. Free-floating sections were incubated for double labeling immunofluorescence at 4 °C with primary antibodies rabbit Beclin-1 (1:100 in PBS, ProSci Inc., Poway, CA, USA), or LC3 (1:100 in PBS, Cell Signaling Technology, Inc., Leiden, The Netherlands), or p62 (1:100 in PBS, Abcam, Cambridge, UK) in combination with mouse anti-NeuN for neurons (1:500 in PBS, Millipore, Temecula, CA, USA). Some sections marked with LC3 were incubated with mouse anti-GFAP for astrocytes (1:5000 in PBS, Merck, Darmstadt, Germany) or mouse anti-CD68 for activated microglia and macrophages (1:100 in PBS, Bio-Rad, Hercules, CA, USA) antibodies. After rinsing, the primary antibodies were detected by exposing sections to appropriate donkey secondary antibodies: Cy3 anti-rabbit (1:400, Jackson Immunoresearch Laboratories, West Grove, PA, USA) and Alexa Fluor 488 anti-mouse (1:250, Jackson Immunoresearch Laboratories, West Grove, PA, USA) for 60 min at RT. Finally, sections were mounted on slides, air-dried, and cover-slipped with a fluoromount aqueous medium (Sigma-Aldrich, St. Louis, MO, USA). 

Immunofluorescence images were analyzed and photographed under a Zeiss LSM 510 confocal laser scanning microscopy system. 

To verify and confirm the specificity of the immunolabeling, primary antisera were omitted, and the sections were incubated only with secondary antibodies. No immunoreactivity was detected. 

Five non-consecutive sections were processed for double-label immunofluorescence from each animal/time point.

### 4.5. Statistical Analysis 

All statistical analyses were performed using the GraphPad software (GraphPad Prism version 6.01 for Windows, GraphPad Software, La Jolla, CA, USA). The quantitative data were expressed as the mean ± SEM. Differences between control and EAE groups were assessed using an unpaired Student *t*-test, assuming the levels of probability * *p* < 0.05, ** *p* < 0.01, and *** *p* < 0.001 as statistically significant.

## Figures and Tables

**Figure 1 ijms-24-13225-f001:**
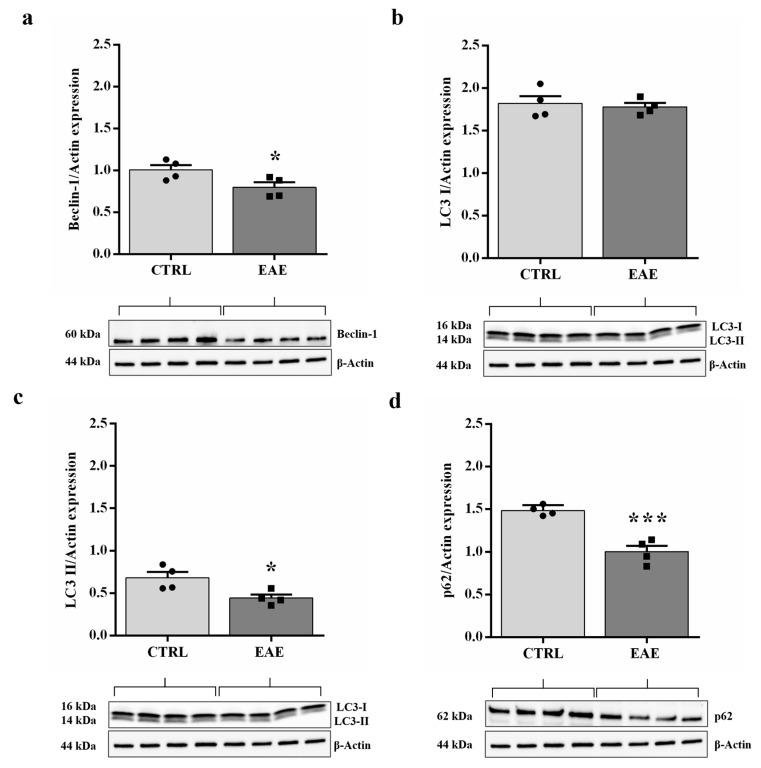
EAE induction reduced Beclin-1, LC3-II, and p62 expression levels and did not affect LC3-I protein in the mouse hippocampus. Graphic representations and Western blot images of Beclin-1 (**a**), LC3 I (**b**), LC3 II (**c**), and p62 (**d**) proteins are shown. Values are displayed as mean ± standard error of the mean (SEM) for each group: the control mice (*n* = 4) and EAE mice (*n* = 4), * *p* < 0.05 and *** *p* < 0.001 compared with controls, Student *t*-test. CTRL: control sample. EAE: Experimental Autoimmune Encephalomyelitis.

**Figure 2 ijms-24-13225-f002:**
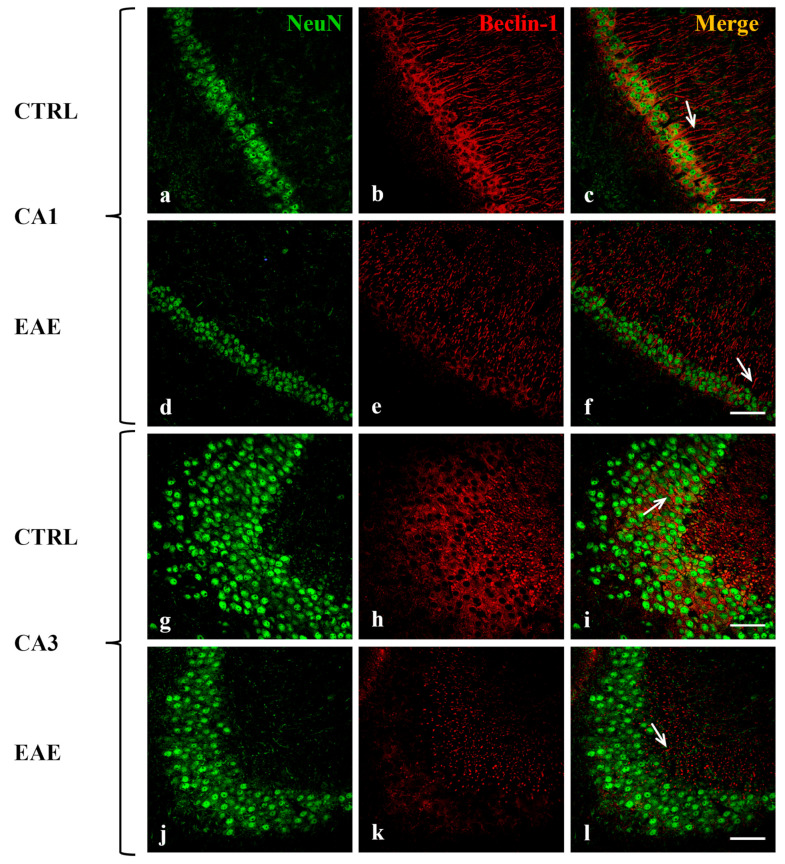
Localization and expression of Beclin-1 in mouse hippocampal neurons of CA1 and CA3 areas during EAE are shown. Sections of the CA1 and CA3 fields are labeled for Neu N (green, **a**,**d**,**g**,**j**), Beclin-1 (red, **b**,**e**,**h**,**k**), and Neu N/Beclin-1 (merge, **c**,**f**,**i**,**l**). Control sections (**a**–**c**,**g**–**i**). Beclin-1 staining in CA1 (**e**) and CA3 (**k**) of the EAE group is more faded than controls. Notably, it is evident a progressive loss of Neu N-positive neurons in EAE mice in CA1 (**d**) and CA3 (**j**) compared with control mice (**a**,**g**). Arrows show details. Scale bar: 50 µm. CTRL: control sample. EAE: Experimental Autoimmune Encephalomyelitis.

**Figure 3 ijms-24-13225-f003:**
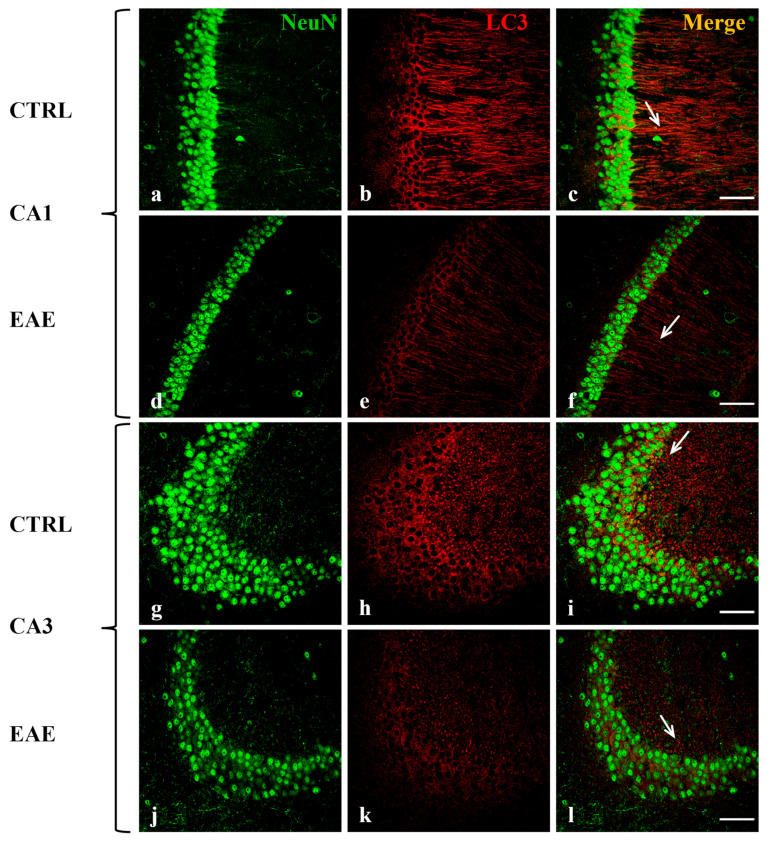
Localization and expression of LC3 in mouse hippocampal neurons of CA1 and CA3 areas during EAE are shown. Sections of the CA1 and CA3 fields are labeled for Neu N (green, **a**,**d**,**g**,**j**), LC3 (red, **b**,**e**,**h**,**k**), and Neu N/LC3 (merge, **c**,**f**,**i**,**l**). Control sections (**a**–**c**,**g**–**i**). LC3 staining is more faded in CA1 (**e**) and CA3 (**k**) of EAE animals compared to controls. Note the progressive loss of Neu N-positive neurons in EAE mice in CA1 (**d**) and CA3 (**j**) compared with control mice (**a**,**g**). Arrows show details. Scale bar: 50 µm. CTRL: control sample. EAE: Experimental Autoimmune Encephalomyelitis.

**Figure 4 ijms-24-13225-f004:**
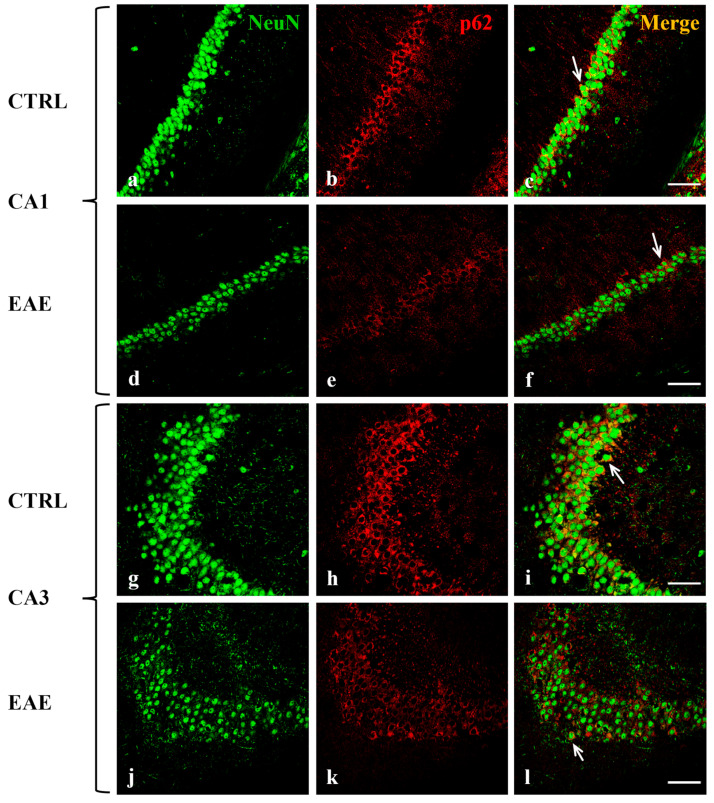
Localization and expression of p62 in mouse hippocampal neurons of CA1 and CA3 areas during EAE are shown. Sections of the CA1 and CA3 fields are labeled for Neu N (green, **a**,**d**,**g**,**j**), p62 (red, **b**,**e**,**h**,**k**), and Neu N/LC3 (merge, **c**,**f**,**i**,**l**). Control sections (**a**–**c**,**g**–**i**). p62 staining decreased in CA1 (**e**) and CA3 (**k**) during EAE. Interestingly, Neu N-positive neurons are less represented in EAE mice in CA1 (**d**) and CA3 (**j**) compared with control mice (**a**,**g**). Arrows show details. Scale bar: 50 µm. CTRL: control sample. EAE: Experimental Autoimmune Encephalomyelitis.

**Figure 5 ijms-24-13225-f005:**
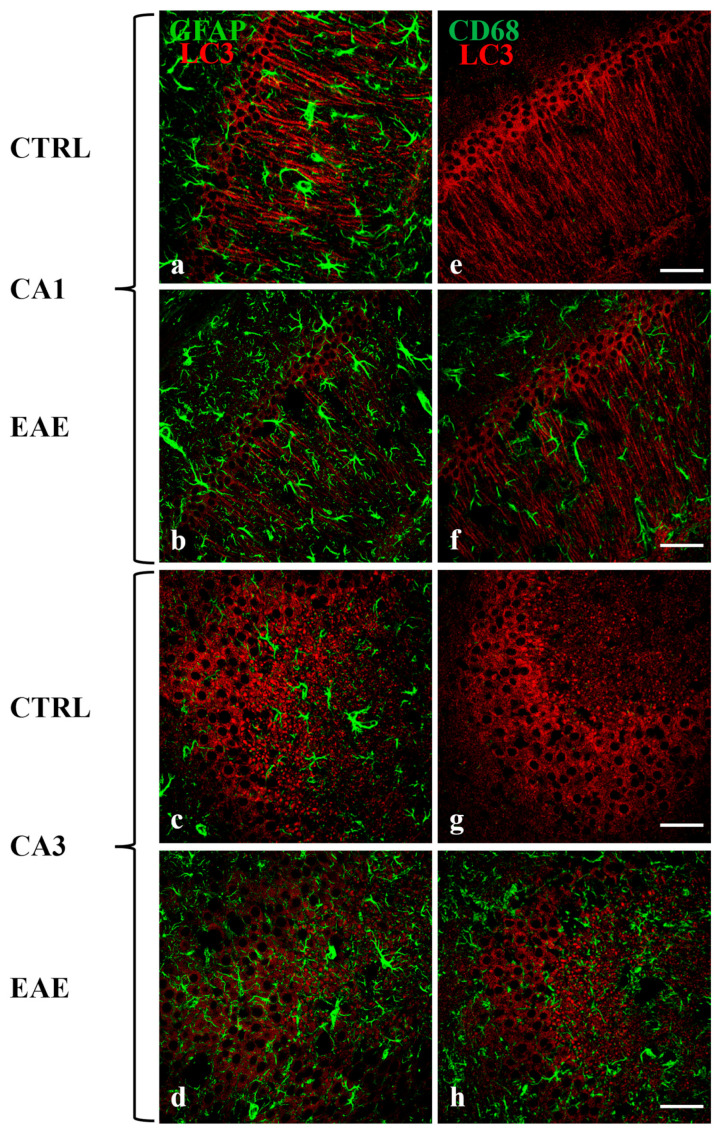
The colocalization of LC3 labeling with the hippocampal astrocytes and activated microglia of CA1 and CA3 areas during EAE was not observed. Sections of the CA1 and CA3 fields of EAE mice are labeled for GFAP/LC3 (green/red, **b**,**d**) and CD68/LC3 (green/red, **f**,**h**). Control sections (**a**,**c**,**e**,**g**). Note the progressive increase in GFAP-positive astrocytes (**b**,**d**) and CD68-positive microglia (**f**,**h**) in EAE mice compared with control mice (**a**,**c**,**e**,**g**). Scale bar: 50 µm. CTRL: control sample. EAE: Experimental Autoimmune Encephalomyelitis.

**Figure 6 ijms-24-13225-f006:**
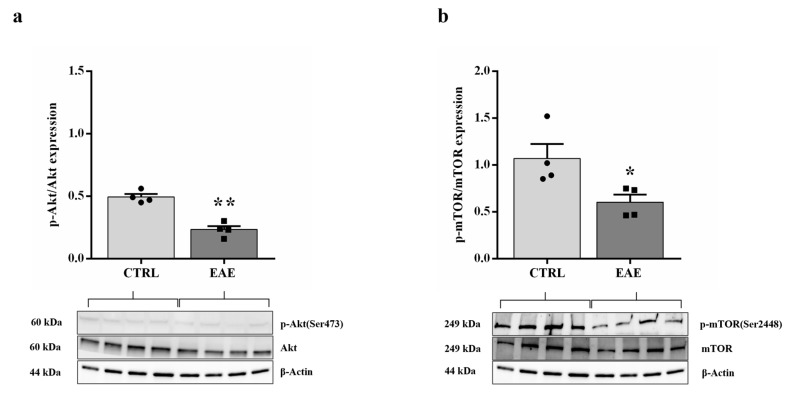
EAE induction reduced expression levels of pAkt and p-mTOR in the mouse hippocampus. Graphic representations and Western blot images of p-Akt(Ser473)/Akt (**a**) and p-mTOR(Ser2448)/mTOR (**b**) are shown. Values are displayed as mean ± standard error of the mean (SEM) for each group: the control mice (*n* = 4) and EAE mice (*n* = 4), * *p* < 0.05 and ** *p* < 0.01 compared with controls. Student *t*-test, CTRL: control sample. EAE: Experimental Autoimmune Encephalomyelitis.

**Figure 7 ijms-24-13225-f007:**
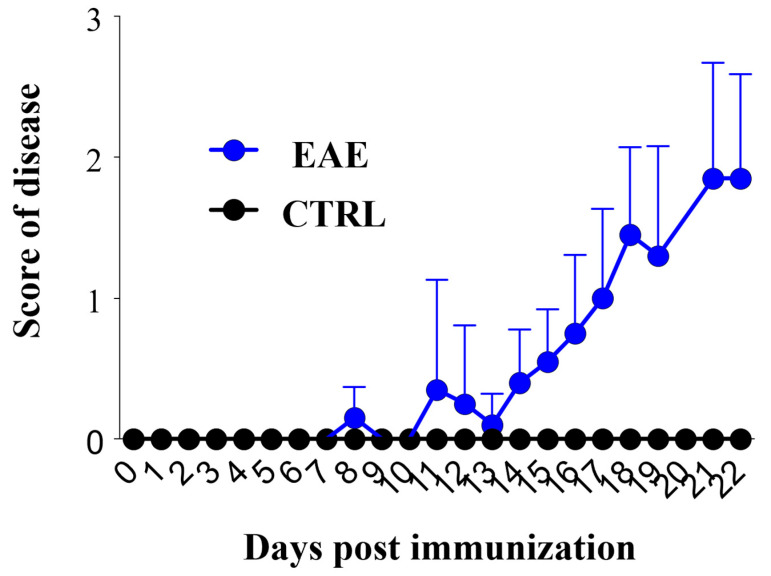
Clinical Score and Symptoms of Experimental Autoimmune Encephalomyelitis (EAE) C57Bl/6 mice. The values of the clinical scores of EAE-affected mice (*n* = 8) and controls (*n* = 8) are displayed in the graph, resulting from the daily clinical evaluation as described in the Materials and Methods section. CTRL: control sample.

## Data Availability

Not applicable.

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
