# Peer review of "Altered Expression of Autophagy Biomarkers in Hippocampal Neurons in a Multiple Sclerosis Animal Model"

_ijms, 2023, doi:10.3390/ijms241713225_

Round 1
Reviewer 1 Report (Previous Reviewer 1)
Authors have significantly revised the manuscript and all concerned queries and suggestions have been satisfactorily incorporated.
Author Response
We thank the Reviewer for positive evaluation of our paper.
Reviewer 2 Report (Previous Reviewer 2)
Dear Authors,
We sincerely appreciate your efforts in revising the manuscript as per the suggestions provided. We have taken note of your clarifications regarding the focus of your study on the hippocampus in the context of autophagy markers and their relevance to EAE. Your response has shed light on the novel perspective you aimed to bring to the field.
I have cited few articles for your kind reference, where the associated researchers have measured the autophagy biomarkers in EAE conditions. Please go through it.
1. Feng X, Hou H, Zou Y, Guo L. "Defective autophagy is associated with neuronal injury in a mouse model of multiple sclerosis." Bosn J Basic Med Sci. 2017 May 20;17(2):95-103. doi: 10.17305/bjbms.2017.1696. PMID: 28086065; PMCID: PMC5474114. This study evaluated levels of LC3-II, Beclin1, and p62 expression.
2. Nutma E, Marzin MC, Cillessen SA, Amor S. "Autophagy in white matter disorders of the CNS: mechanisms and therapeutic opportunities." J Pathol. 2021 Feb;253(2):133-147. doi: 10.1002/path.5576. Epub 2020 Dec 4. PMID: 33135781; PMCID: PMC7839724. This manuscript revealed the role of the AKT/mTOR pathway in autophagy and its involvement in EAE.
3. Wang J, Song X, Tan G, Sun P, Guo L, Zhang N, Wang J, Li B. "NAD+ improved experimental autoimmune encephalomyelitis by regulating SIRT1 to inhibit PI3K/Akt/mTOR signaling pathway." Aging (Albany NY). 2021 Dec 20;13(24):25931-25943. doi: 10.18632/aging.203781. Epub 2021 Dec 20. PMID: 34928817; PMCID: PMC8751589. This manuscript undoubtedly reports the role of the PI3K/Akt/mTOR signalling pathway in EAE-associated autophagy.
We acknowledge that certain aspects of autophagy markers and associated pathways, such as LC3-II, Beclin1, p62, AKT/pAKT and mTOR/ pMTOR, have been explored in previous research related to EAE. We commend your specificity in studying the hippocampus and employing Western blotting (WB) and immunohistochemistry (IHC) techniques to measure expression levels of these markers.
However, it's important to note that the availability of previous research that covers these markers and pathways may impact the perceived novelty of your findings. Our objective as reviewers is to ensure that the research contributes significantly to the existing knowledge base. While your focus on the hippocampus is indeed a unique aspect, the key lies in how you differentiate your study from the already accessible information.
Considering this, we encourage you to further emphasize the distinctiveness of your research. You might consider elaborating on how your study adds a fresh perspective to the role of these markers specifically in the hippocampus in the context of EAE.
We look forward to seeing how you can build upon the foundations you've laid and make your research truly stand out in the field. Your dedication to advancing our understanding of autophagy in the context of EAE is commendable, and by accentuating the unique contributions of your study, you can strengthen its impact and originality.
Thank you for your continued efforts, and we are here to support you in further enhancing the significance of your research.
Best regards,
The Reviewer
Author Response
We would like to thank the Reviewer for the constructive comments, suggestions, and encouragement.
Dear Authors,
We sincerely appreciate your efforts in revising the manuscript as per the suggestions provided. We have taken note of your clarifications regarding the focus of your study on the hippocampus in the context of autophagy markers and their relevance to EAE. Your response has shed light on the novel perspective you aimed to bring to the field.
I have cited few articles for your kind reference, where the associated researchers have measured the autophagy biomarkers in EAE conditions. Please go through it.
- Feng X, Hou H, Zou Y, Guo L. "Defective autophagy is associated with neuronal injury in a mouse model of multiple sclerosis." Bosn J Basic Med Sci. 2017 May 20;17(2):95-103. doi: 10.17305/bjbms.2017.1696. PMID: 28086065; PMCID: PMC5474114. This study evaluated levels of LC3-II, Beclin1, and p62 expression.
- Nutma E, Marzin MC, Cillessen SA, Amor S. "Autophagy in white matter disorders of the CNS: mechanisms and therapeutic opportunities." J Pathol. 2021 Feb;253(2):133-147. doi: 10.1002/path.5576. Epub 2020 Dec 4. PMID: 33135781; PMCID: PMC7839724. This manuscript revealed the role of the AKT/mTOR pathway in autophagy and its involvement in EAE.
- Wang J, Song X, Tan G, Sun P, Guo L, Zhang N, Wang J, Li B. "NAD+ improved experimental autoimmune encephalomyelitis by regulating SIRT1 to inhibit PI3K/Akt/mTOR signaling pathway." Aging (Albany NY). 2021 Dec 20;13(24):25931-25943. doi: 10.18632/aging.203781. Epub 2021 Dec 20. PMID: 34928817; PMCID: PMC8751589. This manuscript undoubtedly reports the role of the PI3K/Akt/mTOR signalling pathway in EAE-associated autophagy.
We acknowledge that certain aspects of autophagy markers and associated pathways, such as LC3-II, Beclin1, p62, AKT/pAKT and mTOR/ pMTOR, have been explored in previous research related to EAE. We commend your specificity in studying the hippocampus and employing Western blotting (WB) and immunohistochemistry (IHC) techniques to measure expression levels of these markers.
However, it's important to note that the availability of previous research that covers these markers and pathways may impact the perceived novelty of your findings. Our objective as reviewers is to ensure that the research contributes significantly to the existing knowledge base. While your focus on the hippocampus is indeed a unique aspect, the key lies in how you differentiate your study from the already accessible information.
Considering this, we encourage you to further emphasize. You might consider elaborating on how your study adds a fresh perspective to the role of these markers specifically in the hippocampus in the context of EAE.
We look forward to seeing how you can build upon the foundations you've laid and make your research truly stand out in the field. Your dedication to advancing our understanding of autophagy in the context of EAE is commendable, and by accentuating the unique contributions of your study, you can strengthen its impact and originality.
According to the requests advanced by the Reviewer, in the Discussion section we added considerations concerning the originality and distinctiveness of results obtained (highlighted in green).
Reviewer 3 Report (New Reviewer)
It is an interesting manuscript, it is presented with a good selection of assays demonstrating expression levels, distribution, and cellular localization of the main autophagy-related proteins (Beclin-1, LC3 and p62) in the hippocampus of experimental autoimmune encephalomyelitis affected mice. Methods are overall clear and appropriate statistics are used.
1. A reference is necessary for the phrase “The results demonstrated that, after EAE-treatment, the levels of p-Akt and p-mTOR were significantly decreased, indicating that autophagy increased via inhibiting Akt/mTOR pathway”.
2. It is necessary to explain in more details the upregulation of autophagy in the EAE-derived hippocampal neurons.
3. Based on the results thus obtained can be proposed the diverse interpretation. The recession of expression levels of Beclin-1 and p62 is a consequence of protein synthesis reduction by means of p-Akt and p-mTOR downregulation.
Author Response
We would like to thank the Reviewer for the constructive comments and suggestions.
#1 A reference is necessary for the phrase “The results demonstrated that, after EAE-treatment, the levels of p-Akt and p-mTOR were significantly decreased, indicating that autophagy increased via inhibiting Akt/mTOR pathway”.
As indicated, we added the reference (highlighted in yellow) in the text of the Discussion, line 364.
#2 It is necessary to explain in more details the upregulation of autophagy in the EAE-derived hippocampal neurons.
As requested, we detailed the upregulation of autophagy in the EAE-affected hippocampal neurons at the end of the Discussion (highlighted in yellow).
#3 Based on the results thus obtained can be proposed the diverse interpretation. The recession of expression levels of Beclin-1 and p62 is a consequence of protein synthesis reduction by means of p-Akt and p-mTOR downregulation.
We thank the Reviewer for this very interesting interpretation of the data. Indeed, the downregulation of p-Akt and p-mTOR could lead to a reduced expression of Beclin-1 (Yu, X. et al., Autophagy 2015, 11, 1711–1728). Instead, the decreased expression levels of p62 are reasonably a direct consequence of activated autophagy. As we described in text: “p62 accumulates when autophagic activity is inhibited and its levels are low when autophagy is activated (Bjørkøy, G. et al., Chapter 12, Methods in Enzymology 2009, 452, 181–197. Larsen, K.B. et al., Autophagy 2010, 6, 784–793), being this protein a selective substrate of autophagy".
Round 2
Reviewer 2 Report (Previous Reviewer 2)
Dear Authors,
I find myself unconvinced by the latest update of the manuscript. Simply presenting a well-known pathway in a different region of the brain, such as the hippocampus, does not inherently qualify as a novel discovery. Additionally, I've noticed several linguistic errors throughout the manuscript.
For the manuscript to be considered "novel," it would have required a demonstration of mechanistic variations within the specified brain region, preferably accompanied by innovative therapeutic interventions. Without such demonstrations, I struggle to perceive any novelty in terms of mechanisms or therapeutic activities. To truly establish novelty, it is essential that the proposed mechanisms be substantiated using unique therapeutic approaches.
Kindly take these suggestions into account for further revisions.
Sincerely,
The Reviewer
Please improve the language.
This manuscript is a resubmission of an earlier submission. The following is a list of the peer review reports and author responses from that submission.
Round 1
Reviewer 1 Report
The present article by Ceccariglia et al. entitled "Altered Expression of Autophagy Biomarkers in Hippocampal Neurons in a Multiple Sclerosis Animal Model" demonstrates the evaluation of autophagy mechanism in hippocampal neurons of an EAE induced MS mouse model. Authors have evaluated the levels of different autophagy markers including Beclin-1, LC3 and p62 and Akt/mTOR pathway in hippocampal neurons of this MS model by using biochemical techniques, namely western blotting and immunohistochemistry. There are several queries/ comments that need to be addressed by authors as follows:
Query#1; How did authors optimize the dosage and duration of administering EAE to mice?
Query#2; Why did authors used female mice to induce MS using EAE? Is there any specific rationale?
Query#3; In figure 7, authors represented scoring of disease on Y-axis. This is very vague and hard to understand. Please elaborate the paradigms used in this study for clinical evaluation of MS in this specific model.
Query#4; In figure 1, authors need to show the bar graph representing individual plots of each animals (n=5). Full western blot image needs to be shown. Also, LC3 II/I ratio bar graph is missing which is very critical to evaluate autophagy.
Query#5; In figure 2, immunofluorescence image scale bars need to be added in each image. fig. 2a-c and 2d-f seems to be of different magnification that is not acceptable to comparing the fluorescence signals. High resolution confocal images (40x magnification) is needed to evaluate Beclin-1 specific staining. Quantification of fluorescence intensity is missing which is very important for comparing control vs EAE mice.
Query#6; In figure 3 and 4, authors need to address similar issues as mentioned in query#5. For LC3 and p62 immunostaining, authors must show punctate immunostaining using high resolution confocal images. Quantification of punta is also needed to draw any conclusion in this study.
Query#7; In figure 5, why authors did not show Iba-I staining for microglia?
Query#8; In figure 6, authors need to show full western blot images and revise bar graphs representing individual animals.
Comment: thorough revision of english language is needed.
Query#6:
English language needs thorough editing and revision throughout the manuscript.
Author Response
We would like to thank the Reviewer for the constructive comments and suggestions.
#1 How did authors optimize the dosage and duration of administering EAE to mice?
The procedures to immunologically induce EAE, a model of encephalomyelitis that mimic different demyelinating disorders, as MS, are standardized and may change depending on different variables (mouse strain, type of demyelinating disorder, needs to synchronize clinical symptoms, distribution of the lesions and other main features of EAE). In our paper we used a well described and recognized murine model of Multiple Sclerosis, obtained immunizing C57Bl/6 mice with a peptide derived from myelin, Myelin oligodendrocyte glycoprotein (MOG). The dosage to immunize animals and to actively induce EAE should be comprised between 10 and 300 microg/mouse (Stromnes IM, Goverman JM. Active induction of experimental allergic encephalomyelitis. Nat Protoc. 2006;1(4):1810-9. doi: 10.1038/nprot.2006.285). We opted for a low concentration of antigen because in our experience high amount of antigen in the emulsion could generate an anergic immune response more often than low doses of antigen.
#2 Why did authors used female mice to induce MS using EAE? Is there any specific rationale?
We thank the reviewer for this very interesting question. The gender differences on EAE model are often debated. In our experience with this specific and other EAE model that we had the opportunity to manage in the last 20 years (C57Bl/6, SJL/J, Biozzi, DEREG, vb10KO and S100BKO mice) more than 90% of immunized females were able to develop the disease and as showed in our previous works, we found homogenous results regarding distribution of the lesions and synchronization of symptoms. In male the results are often heterogenous, causing issues about reduction of the number of animals. Moreover, an important component of EAE is the sociality of individual. It is fundamental that animals can be housed in cages containing at least three mice, otherwise the course of the disease could be compromised. This is simple with females, while aggressive males could impact on this aspect and generate biases such as injuries or impairments of cage mates that could then influence clinical visits and in general also the 3Rs rules. Finally, the big part of EAE mouse models in the literature used females more often than males also for the fact that autoimmune diseases in humans, such as Multiple Sclerosis, are more frequent in female than in male.
#3 In figure 7, authors represented scoring of disease on Y-axis. This is very vague and hard to understand. Please elaborate the paradigms used in this study for clinical evaluation of MS in this specific model.
As requested, we detailed clinical evaluation and scoring procedures in red at the end of the methods chapter 4.1.
#4 In figure 1, authors need to show the bar graph representing individual plots of each animal (n=5). Full western blot image needs to be shown. Also, LC3 II/I ratio bar graph is missing which is very critical to evaluate autophagy.
As suggested by the reviewer, in figure 1, bar graph representing individual plots of each animal is added in each graphic. The number of animals was originally n=5. For problems of method of homogenization with a hippocampal sample, n=4 has been used for the analysis of western blot, corresponding to the bands showed in figure. We emendated the number of animals accordingly.
#5 In figure 2, immunofluorescence image scale bars need to be added in each image. fig. 2a-c and 2d-f seems to be of different magnification that is not acceptable to comparing the fluorescence signals. High resolution confocal images (40x magnification) is needed to evaluate Beclin-1 specific staining. Quantification of fluorescence intensity is missing which is very important for comparing control vs EAE mice.
Following the suggestion of the reviewer, in figure 2 we have added scale bar in each immunofluorescence merge image. The images involving markers, NeuN and Beclin-1, of control and EAE animals of the CA1 and CA3 hippocampal regions were photographed at the same magnification as the merges, therefore the scale bar is the same and it is not added.
The images of figure 2, included 2a-c and 2d-f, were all photographed under the same magnification (40x).
Quantification of fluorescence intensity in the different hippocampal areas would have determined a partial analysis both of Beclin-1 and other autophagic protein expression. Hippocampus, indeed, shows regional differences in the expression of Beclin-1 and all the other markers analyzed. No evident labeling was detected and observed by confocal microscopy in CA4 subfield and Dentate Gyrus (as described on page 9 lines 251-252), probably because the low concentration levels of protein in this area was poorly detectable with our low sensitive microscopy method. To complete these results, we analyzed the total amount of protein in the homogenates of the whole hippocampus with western blot assay as described in Materials and Methods.
#6 In figure 3 and 4, authors need to address similar issues as mentioned in query#5. For LC3 and p62 immunostaining, authors must show punctate immunostaining using high resolution confocal images. Quantification of punta is also needed to draw any conclusion in this study.
In figures 3 and 4 we have added scale bar in each immunofluorescence merge image, but not in the images of single markers as mentioned in the answer to question 5.
Regarding LC3 and p62 punctate immunostaining, unfortunately, we could not observe punctate fluorescence of these markers at 40X, most likely for the method of processing the tissue samples and the thickness of the sections used for microscopy analysis. Anyhow, LC3 punctate corresponds to the LC3-II active form (correlated to the number of vacuoles inside the cell), as indicated on page 3 lines 112-114. So, the amount of LC3-II can be used to monitor autophagosome formation (Koike M et al., 2005), and it can be easily detected by western blot analysis quantifying the LC3-II band (14 kDa) corresponding to the protein total expression in the whole hippocampus. Regarding p62, western blot assay can easily be used to monitor autophagy activation through analysis of the change in p62 amounts as described by Klionsky D.J. et al., 2008. Indeed, “p62 amount accumulates when autophagic activity is inhibited and it decreases when autophagy is activated”, as described on page 13 lines 467-468.
For fluorescence quantification of LC3 and p62 please refer to the answer to question 5.
#7 In figure 5, why authors did not show Iba-I staining for microglia?
Iba-1 microglia marker, generally, is more used for structural studies in absence of pathology. CD68, on the other hand, reflects immune activation and responses to damaged tissue, therefore, it is the marker more suitable to detect (activated) microglia in the animal model of Multiple Sclerosis we used in this study.
#8 In figure 6, authors need to show full western blot images and revise bar graphs representing individual animals.
Please refer to the answer to question 4.
Comment: thorough revision of english language is needed.
We revised English language in track change modality.
Reviewer 2 Report
Dear authors,
Your research lacks novelty in the field of EAE associated autophagy markers, as the information regarding the upregulation and downregulation of these markers has already been published and is readily accessible in the public domain. This suggests that the findings of your research may not contribute significantly to the existing knowledge base.
To make your research stand out, it is important to highlight how it differs from the already available information. By exploring unique aspects or uncovering new insights related to autophagy markers, you can provide a fresh perspective on this topic. This could involve investigating previously unexplored pathways, identifying novel markers, or examining the role of autophagy in specific disease contexts. By emphasizing these distinctions, you can demonstrate the originality and value of your research.
Furthermore, it is crucial to go beyond merely presenting the information about autophagy markers and their expression levels. Without establishing a clear connection to potential therapeutic effects, your research may lack practical significance. Consider how your findings can be translated into meaningful clinical applications or contribute to the development of targeted therapies. By providing context and discussing the potential implications of your research on therapeutic interventions or disease management, you can highlight its relevance and significance in the field.
In summary, to enhance the significance of your research, focus on differentiating it from existing information, exploring novel aspects within the field of autophagy markers, and establishing its potential therapeutic implications. This will strengthen the originality, value, and impact of your research contribution.
Thanks,
The Reviewer
Author Response
We would like to thank the Reviewer for the constructive comments and suggestions.
Dear authors,
Your research lacks novelty in the field of EAE associated autophagy markers, as the information regarding the upregulation and downregulation of these markers has already been published and is readily accessible in the public domain. This suggests that the findings of your research may not contribute significantly to the existing knowledge base.
To make your research stand out, it is important to highlight how it differs from the already available information. By exploring unique aspects or uncovering new insights related to autophagy markers, you can provide a fresh perspective on this topic. This could involve investigating previously unexplored pathways, identifying novel markers, or examining the role of autophagy in specific disease contexts. By emphasizing these distinctions, you can demonstrate the originality and value of your research.
Furthermore, it is crucial to go beyond merely presenting the information about autophagy markers and their expression levels. Without establishing a clear connection to potential therapeutic effects, your research may lack practical significance. Consider how your findings can be translated into meaningful clinical applications or contribute to the development of targeted therapies. By providing context and discussing the potential implications of your research on therapeutic interventions or disease management, you can highlight its relevance and significance in the field.
In summary, to enhance the significance of your research, focus on differentiating it from existing information, exploring novel aspects within the field of autophagy markers, and establishing its potential therapeutic implications. This will strengthen the originality, value, and impact of your research contribution.
Thanks,
Autophagy involvement in MS and in EAE animal model has been mainly studied, as it appears to result in the scientific literature, in the immune system, and in the central nervous system, especially in the spinal cord which is the region primarily involved in the disease. In the brain, the hippocampus is known to play a crucial role at the basis relevant symptoms (10,14,50-54 in the manuscript), although cellular and molecular mechanisms underlying these symptoms, including in particular autophagic processes and biomarkers, have not been definitely investigated. Thus, the study of autophagy biomarkers in the hippocampus of mice affected by a recognized model of MS (EAE) constitutes an element of novelty, which is in fact present in our work, and that now we more clearly evidenced in the present version of the revised manuscript (in green).
Essentially, through this study, we wanted to provide a basis for elucidation of autophagy activation in the hippocampus, which may constitute an important player of MS processes, which appears not to be adequately investigated, and we started, necessarily, from analysis of the primary molecular mechanisms, which constitute a prerequisite for further studies more deeply investigating these processes, as we indicated more clearly in the revised version of the manuscript (in green).
Thus, the goal of this study does not reside in finding novel aspects of autophagy in general terms, which would be a very wide goal, possibly regarding the majority of pathogenic processes, and which goes beyond our intents. Merely we identified in the hippocampus of mice affected by a recognized experimental model of MS (EAE) some hallmarks of autophagy, constituted by alterations in recognized autophagy biomarkers, which had never previously investigated in EAE hippocampus and which, as a consequence, propose autophagy processes as a part of EAE pathogenic processes in the hippocampus, which appear not yet having been extensively investigated.
Finally, according to the requests advanced by the Reviewer, in the Discussion section we added considerations concerning the potential therapeutic implications of results obtained (in red).